# Self-supervised Learning from a Multi-view Perspective

**Yao-Hung Hubert Tsai, Yue Wu, Ruslan Salakhutdinov, Louis-Philippe Morency**
Machine Learning Department, Carnegie Mellon University

## Abstract

As a subset of unsupervised representation learning, self-supervised representation learning adopts self-defined signals as supervision and uses the learned representation for downstream tasks, such as object detection and image captioning. Many proposed approaches for self-supervised learning follow naturally a multi-view perspective, where the input (e.g., original images) and the self-supervised signals (e.g., augmented images) can be seen as two redundant views of the data. Building from this multi-view perspective, this paper provides an information-theoretical framework to better understand the properties that encourage successful self-supervised learning. Specifically, we demonstrate that self-supervised learned representations can extract task-relevant information and discard task-irrelevant information. Our theoretical framework paves the way to a larger space of self-supervised learning objective design. In particular, we propose a composite objective that bridges the gap between prior contrastive and predictive learning objectives, and introduce an additional objective term to discard task-irrelevant information. To verify our analysis, we conduct controlled experiments to evaluate the impact of the composite objectives. We also explore our framework's empirical generalization beyond the multi-view perspective, where the cross-view redundancy may not be clearly observed.

## 1 Introduction

Self-supervised learning (SSL) (Zhang et al., 2016; Devlin et al., 2018; Oord et al., 2018; Tian et al., 2019) learns representations using a proxy objective (i.e., SSL objective) between inputs and self-defined signals. Empirical evidence suggests that the learned representations can generalize well to a wide range of downstream tasks, even when the SSL objective has not utilize any downstream supervision during training. For example, SimCLR (Chen et al., 2020) defines a contrastive loss (i.e., an SSL objective) between images with different augmentations (i.e., one as the input and the other as the self-supervised signal). Then, one can take SimCLR as features extractor and adopt the features to various computer vision applications, spanning image classification, object detection, instance segmentation, and pose estimation (He et al., 2019). Despite success in practice, only a few work (Arora et al., 2019; Lee et al., 2020; Tosh et al., 2020) provide theoretical insights into the learning efficacy of SSL. Our work shares a similar goal to explain the success of SSL, from the perspectives of Information Theory (Cover & Thomas, 2012) and multi-view representation[1].

To understand (a subset[2] of) SSL, we start by the following *multi-view assumption*. First, we regard the input and the self-supervised signals as two corresponding views of the data. Using our running example, in SimCLR (Chen et al., 2020), the augmented images (i.e., the input and the self-supervised signal) are an image with different views. Second, we adopt a common assumption in multi-view learning: either view alone is (approximately) sufficient for the downstream tasks (see Assumption 1 in prior work (Sridharan & Kakade, 2008)). The assumption suggests that the image augmentations (e.g., changing the style of an image) should not affect the labels of images, or analogously, the self-supervised signal contains most (if not all) of the information that the input has about the downstream tasks. With this assumption, our first contribution is to formally show that the self-supervised learned

---

[1]The work (Lee et al., 2020; Tosh et al., 2020) are done concurrent and in parallel, and part of their assumptions/ conclusions are similar to ours. We will elaborate the differences more in the related work section.

[2]We discuss the limitations of the multi-view assumption in Section 2.1.

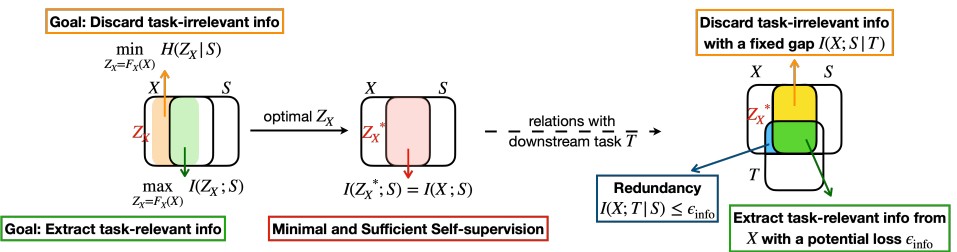

Figure 1: High-level takeaways for our main results using information diagrams. (a) We present to learn minimal and sufficient self-supervision: minimize $H(Z_X|S)$ for discarding task-irrelevant information and maximize $I(Z_X; S)$ for extracting task-relevant information. (b) The resulting learned representation $Z_X^*$ contains all task relevant information from the input with a potential loss $\epsilon_{\mathrm{info}}$ and discards task-irrelevant information with a fixed gap $I(X; S|T)$. (c) Our core assumption: the self-supervised signal is approximately redundant to the input for the task-relevant information.

representations can 1) extract all the task-relevant information (from the input) with a potential loss; and 2) discard all the task-irrelevant information (from the input) with a fixed gap. Then, using classification task as an example, we are able the quantify the smallest generalization error (Bayes error rate) given the discussed task-relevant and task-irrelevant information.

As the second contribution, our analysis 1) connects prior arts for SSL on contrastive (Oord et al., 2018; Bachman et al., 2019; Chen et al., 2020; Tian et al., 2019) and predictive learning (Zhang et al., 2016; Vondrick et al., 2016; Tulyakov et al., 2018; Devlin et al., 2018) approaches; and 2) paves the way to a larger space of composing SSL objectives to extract task-relevant and discard task-irrelevant information simultaneously. For instance, the combination between the contrastive and predictive learning approaches achieves better performance than contrastive- or predictive-alone objective and enjoys less over-fitting problem. We also present a new objective to discard task-irrelevant information. The objective can be easily incorporated with prior self-supervised learning objectives.

We conduct controlled experiments on visual (the first set) and visual-textual (the second set) self-supervised representation learning. The first set of experiments are performed when the multi-view assumption is likely to hold. The goal is to compare different compositions of SSL objectives on extracting task-relevant and discarding task-irrelevant information. The second set of experiments are performed when the input and the self-supervised signal lie in very different modalities. Under this cross-modality setting, the task-relevant information may not mostly lie in the shared information between the input and the self-supervised signal. The goal is to examine SSL objectives' generalization, where the multi-view assumption is likely to fail.

## 2  A MULTI-VIEW INFORMATION-THEORETICAL FRAMEWORK

**Notations.**  For the input, we denote its random variable as $X$, sample space as $\mathcal{X}$, and outcome as $x$. We learn a representation ($Z_X$/ $\mathcal{Z}$/ $z_x$) from the input through a deterministic mapping $F_X$: $Z_X = F_X(X)$. For the self-supervised signal, we denote its random variable/ sample space/ outcome as $S$/ $\mathcal{S}$/ $s$. Two sample spaces can be different between the input and the self-supervised signal: $\mathcal{X} \neq \mathcal{S}$. The information required for downstream tasks is referred to as "task-relevant information": $T$/ $\mathcal{T}$/ $t$. Note that SSL has no access to the task-relevant information. Lastly, we use $I(A; B)$ to represent mutual information, $I(A; B|C)$ to represent conditional mutual information, $H(A)$ to represent the entropy, and $H(A|B)$ to represent conditional entropy for random variables $A/B/C$. We provide high-level takeaways for our main results in Figure 1. We defer all proofs to Supplementary.

### 2.1  MULTI-VIEW ASSUMPTION

In our paper, we regard the input ($X$) and the self-supervised signals ($S$) as two views of the data. Here, we provide a table showing different $X/S$ in various SSL frameworks:

| Framework | BERT (Devlin et al., 2018) | Look & Listen (Arandjelovic & Zisserman, 2017) | SimCLR (Chen et al., 2020) | Colorization (Zhang et al., 2016) |
|---|---|---|---|---|
| Inputs ($X$) | Non-masked Words | Image | Image | Image Lightness |
| Self-supervised Signals ($S$) | Masked Words | Audio Stream | Same Image with Augmentation | Image Color |

We note that not all SSL frameworks realize the inputs and the self-supervised signals as corresponding views. For instance, Jigsaw puzzle (Noroozi & Favaro, 2016) considers (shuffled) image patches as the input and the positions of the patches as the self-supervised signals. Another example is Learning by Predicting Rotations (Gidaris et al., 2018), which considers an image (rotating with a specific

angle) as the input and the rotation angle of the image as the self-supervised signal. We point out that the frameworks that regard $X/S$ as two corresponding views (Chen et al.; 2020; He et al., 2019) have a much better empirical downstream performance than the frameworks that do not (Noroozi & Favaro, 2016; Gidaris et al., 2018). Our paper hence focuses on the multi-view setting between $X/S$.

Next, we adopt the common assumption (i.e., *multi-view assumption* (Sridharan & Kakade, 2008; Xu et al., 2013)) in the multi-view learning between the input and the self-supervised signal:

**Assumption 1** (Multi-view, restating Assumption 1 in prior work (Sridharan & Kakade, 2008)). *The self-supervised signal is approximately redundant to the input for the task-relevant information. In other words, there exist an $\epsilon_{\text{info}} > 0$ such that $I(X;T|S) \leq \epsilon_{\text{info}}$.*

Assumption 1 states that, when $\epsilon_{\text{info}}$ is small, the task-relevant information lies mostly in the shared information between the input and the self-supervised signals. We argue this assumption is mild with the following example. For self-supervised visual contrastive learning (Hjelm et al., 2018; Chen et al., 2020), the input and the self-supervised signal are the same image with different augmentations. Using image augmentations can be seen as changing the style of an image while not affecting the content. And we argue that the information required for downstream tasks should only be retained in the content but not the style. Next, we point out the failure cases of the assumption (or have large $\epsilon_{\text{info}}$): the input and the self-supervised signal contain very different task-relevant information. For instance, a drastic image augmentation (e.g., adding large noise) may change the content of the image (e.g., the noise completely occludes the objects). Another example is BERT (Devlin et al., 2018), with too much masking, downstream information may exist differently in the masked (i.e., the self-supervised signals) and the non-masked (i.e., the input) words. Analogously, too much masking makes the non-masked words have insufficient context to predict the masked words.

## 2.2 Learning Minimal and Sufficient Representations for Self-supervision

We start by discussing the supervised representation learning. The Information Bottleneck (IB) method (Tishby et al., 2000; Achille & Soatto, 2018) generalizes minimal sufficient statistics to the representations that are minimal (i.e., less complexity) and sufficient (i.e., better fidelity). To learn such representations for downstream supervision, we consider the following objectives:

**Definition 1** (Minimal and Sufficient Representations for Downstream Supervision). Let $Z_X^{\text{sup}}$ be the sufficient supervised representation and $Z_X^{\text{supmin}}$ be the minimal and sufficient representation:

$$Z_X^{\text{sup}} = \arg\max_{Z_X} I(Z_X;T) \text{ and } Z_X^{\text{supmin}} = \arg\min_{Z_X} H(Z_X|T) \text{ s.t. } I(Z_X;T) \text{ is maximized.}$$

To reduce the complexity of the representation $Z_X$, the prior methods (Tishby et al., 2000; Achille & Soatto, 2018) presented to minimize $I(Z_X;X)$ while ours presents to minimize $H(Z_X|T)$. We provide a justification: minimizing $H(Z_X|T)$ reduces the randomness from $T$ to $Z_X$, and the randomness is regarded as a form of incompressibility (Calude, 2013). Hence, minimizing $H(Z_X|T)$ leads to a more compressed representation (discarding redundant information)[3]. Note that we do not constrain the downstream task $T$ as classification, regression, or clustering.

Then, we present SSL objectives to learn sufficient (and minimal) representations for self-supervision:

**Definition 2** (Minimal and Sufficient Representations for Self-supervision). Let $Z_X^{\text{ssl}}$ be the sufficient self-supervised representation and $Z_X^{\text{sslmin}}$ be the minimal and sufficient representation:

$$Z_X^{\text{ssl}} = \arg\max_{Z_X} I(Z_X;S) \text{ and } Z_X^{\text{sslmin}} = \arg\min_{Z_X} H(Z_X|S) \text{ s.t. } I(Z_X;S) \text{ is maximized.}$$

Definition 2 defines our self-supervised representation learning strategy. Now, we are ready to associate the supervised and self-supervised learned representations:

**Theorem 1** (Task-relevant information with a potential loss $\epsilon_{\text{info}}$). *The supervised learned representations (i.e., $Z_X^{\text{sup}}$ and $Z_X^{\text{supmin}}$) contain all the task-relevant information in the input (i.e., $I(X;T)$). The self-supervised learned representations (i.e., $Z_X^{\text{ssl}}$ and $Z_X^{\text{sslmin}}$) contain all the task-relevant information in the input with a potential loss $\epsilon_{\text{info}}$. Formally,*

$$I(X;T) = I(Z_X^{\text{sup}};T) = I(Z_X^{\text{supmin}};T) \geq I(Z_X^{\text{ssl}};T) \geq I(Z_X^{\text{sslmin}};T) \geq I(X;T) - \epsilon_{\text{info}}.$$

---

[3]We do not claim $H(Z_X|T)$ minimization is better than $I(Z_X;X)$ minimization for reducing the complexity in the representations $Z_X$. In Supplementary, we will show that $H(Z_X|T)$ minimization and $I(Z_X;X)$ minimization are interchangeable under our framework's setting.

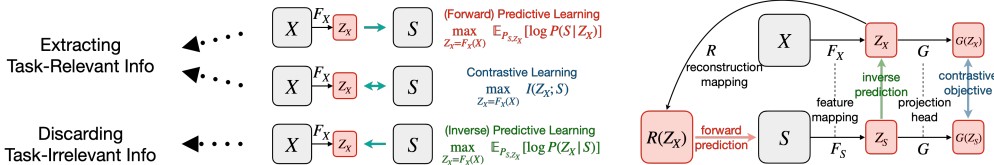

Figure 2: Remarks on contrastive and predictive learning objectives for self-supervised learning. Between the representation $Z_X$ and the self-supervised signal $S$, *contrastive objective* performs mutual information maximization and *predictive objectives* perform log conditional likelihood maximization. We show that the SSL objectives aim at extracting task-relevant and discarding task-irrelevant information. Last, we summarize the computational blocks for practical deployments for these objectives.

When $\epsilon_{\text{info}}$ is small, Theorem 1 indicates that the self-supervised learned representations can extract almost as much task-relevant information as the supervised one. While when $\epsilon_{\text{info}}$ is non-trivial, the learned representations may not always lead to good downstream performance. This result has also been observed in prior work (Tschannen et al., 2019) and InfoMin (Tian et al., 2020), which claim the representations with maximal mutual information may not have the best performance.

**Theorem 2** (Task-irrelevant information with a fixed compression gap $I(X; S|T)$)**.** *The sufficient self-supervised representation (i.e., $I(Z_X^{\text{ssl}}; T)$) contains more task-irrelevant information in the input than the sufficient and minimal self-supervised representation (i.e., $I(Z_X^{\text{ssl}_{\min}}; T)$). The latter contains an amount of the information, $I(X; S|T)$, that cannot be discarded from the input. Formally,*

$$I(Z_X^{\text{ssl}}; X|T) = I(X; S|T) + I(Z_X^{\text{ssl}}; X|S, T) \geq I(Z_X^{\text{ssl}_{\min}}; X|T) = I(X; S|T) \geq I(Z_X^{\text{sup}_{\min}}; X|T) = 0.$$

Theorem 2 indicates that a compression gap (i.e., $I(X; S|T)$) exists when we discard the task-irrelevant information from the input. To be specific, $I(X; S|T)$ is the amount of the shared information between the input and the self-supervised signal excluding the task-relevant information. Hence, $I(X; S|T)$ would be large if the downstream tasks requires only a portion of the shared information.

## 2.3 CONNECTIONS WITH CONTRASTIVE AND PREDICTIVE LEARNING OBJECTIVES

Theorem 1 and 2 state that our self-supervised learning strategies (i.e., $\min H(Z_X|S)$ and $\max I(Z_X; S)$ defined in Definition 2) can extract task-relevant and discard task-irrelevant information. A question emerges:

> "*What are the practical aspects of the presented self-supervised learning strategies?*"

To answer this question, we present 1) the connections with prior SSL objectives, especially for contrastive (Oord et al., 2018; Bachman et al., 2019; Chen et al., 2020; Tian et al., 2019; Hjelm et al., 2018; He et al., 2019) and predictive (Zhang et al., 2016; Pathak et al., 2016; Vondrick et al., 2016; Tulyakov et al., 2018; Peters et al., 2018; Devlin et al., 2018) learning objectives, showing that these objectives are extracting task-relevant information; and 2) a new *inverse predictive learning* objective to discard task-irrelevant information. We illustrate important remarks in Figure 2.

**Contrastive Learning (is extracting task-relevant information).** Contrastive learning objective (Oord et al., 2018) maximizes the dependency/contrastiveness between the learned representation $Z_X$ and the self-supervised signal $S$, which suggests maximizing the the mutual information $I(Z_X; S)$. Theorem 1 suggests that maximizing $I(Z_X; S)$ results in $Z_X$ containing (approximately) all the information required for the downstream tasks from the input $X$. To deploy the contrastive learning objective, we suggest contrastive predictive coding (CPC) (Oord et al., 2018)[4], which is a mutual information lower bound with low variance (Poole et al., 2019; Song & Ermon, 2019):

$$\mathrm{L}_{CL} := \max_{\substack{Z_S = F_S(S), \\ Z_X = F_X(X), G}} \mathbb{E}_{(z_{s_1}, z_{x_1}), \cdots, (z_{s_n}, z_{x_n}) \sim P^n(Z_S, Z_X)} \left[ \frac{1}{n} \sum_{i=1}^{n} \log \frac{e^{\langle G(z_{x_i}), G(z_{s_i}) \rangle}}{\frac{1}{n} \sum_{j=1}^{n} e^{\langle G(z_{x_i}), G(z_{s_j}) \rangle}} \right], \quad (1)$$

where $F_S : \mathcal{S} \rightarrow \mathcal{Z}$ is a deterministic mapping and $G$ is a project head that projects a representation in $\mathcal{Z}$ into a lower-dimensional vector. If the input and self-supervised signals share the same

---

[4]Other contrastive learning objectives can be other mutual information lower bounds such as DV-bound or NWJ-bound (Belghazi et al., 2018) or its JS-divergence (Poole et al., 2019; Hjelm et al., 2018) variants. Among different objectives, Tschannen et al. (2019) have suggested that the objectives with large variance (e.g., DV-/NWJ-bound (Belghazi et al., 2018)) may lead to worsen performance compared to the low variance counterparts (e.g., CPC (Oord et al., 2018) and JS-div. (Poole et al., 2019)).

sample space, i.e., $\mathcal{X} = \mathcal{S}$, we can impose $F_X = F_S$ (e.g., self-supervised visual representation learning (Chen et al., 2020)). The projection head, $G$, can be an identity, a linear, or a non-linear mapping. Last, we note that modeling equation 1 often requires a large batch size (e.g., large $n$ in equation 1) to ensure a good downstream performance (He et al., 2019; Chen et al., 2020).

**Forward Predictive Learning (is extracting task-relevant information).** Forward predictive learning encourages the learned representation $Z_X$ to reconstruct the self-supervised signal $S$, which suggests maximizing the log conditional likelihood $\mathbb{E}_{P_{S,Z_X}}[\log P(S|Z_X)]$. By the chain rule, $I(Z_X; S) = H(S) - H(S|Z_X)$, where $H(S)$ is irrelevant to $Z_X$. Hence, maximizing $I(Z_X; S)$ is equivalent to maximizing $-H(S|Z_X) = \mathbb{E}_{P_{S,Z_X}}[\log P(S|Z_X)]$, which is the predictive learning objective. Together with Theorem 1, if $z_x$ can perfectly reconstruct $s$ for any $(s, z_x) \sim P_{S,Z_X}$, then $Z_X$ contains (approximately) all the information required for the downstream tasks from the input $X$. A common approach to avoid intractability in computing $\mathbb{E}_{P_{S,Z_X}}[\log P(S|Z_X)]$ is assuming a variational distribution $Q_\phi(S|Z_X)$ with $\phi$ representing the parameters in $Q_\phi(\cdot|\cdot)$. Specifically, we present to maximize $\mathbb{E}_{P_{S,Z_X}}[\log Q_\phi(S|Z_X)]$, which is a lower bound of $\mathbb{E}_{P_{S,Z_X}}[\log P(S|Z_X)]$[5]. $Q_\phi(\cdot|\cdot)$ can be any distribution such as Gaussian or Laplacian and $\phi$ can be a linear model, a kernel method, or a neural network. Note that the choice of the reconstruction type of loss depends on the distribution type of $Q_\phi(\cdot|\cdot)$, and is not fixed. For instance, if we let $Q_\phi(S|Z_X)$ be Gaussian $\mathcal{N}\left(S|R(Z_X), \sigma\mathbf{I}\right)$ with $\sigma\mathbf{I}$ as a diagonal matrix[6], the objective becomes:

$$\mathrm{L}_{FP} := \max_{Z_X = F_X(X), R} \mathbb{E}_{s, z_x \sim P_{S,Z_X}}\left[ -\|s - R(z_x)\|_2^2 \right], \tag{2}$$

where $R : \mathcal{Z} \to \mathcal{S}$ is a deterministic mapping to reconstruct $S$ from $Z$ and we ignore the constants derived from the Gaussian distribution. Last, in most real-world applications, the self-supervised signal $S$ has a much higher dimension (e.g., a $224 \times 224 \times 3$ image) than the representation $Z_X$ (e.g., a $64$-dim. vector). Hence, modeling a conditional generative model $Q_\phi(S|Z_X)$ will be challenging.

**Inverse Predictive Learning (is discarding task-irrelevant information).** Inverse predictive learning encourages the self-supervised signal $S$ to reconstruct the learned representation $Z_X$, which suggests maximizing the log conditional likelihood $\mathbb{E}_{P_{S,Z_X}}[\log P(Z_X|S)]$. Given Theorem 2 together with $-H(Z_X|S) = \mathbb{E}_{P_{S,Z_X}}[\log P(Z_X|S)]$, we know if $s$ can perfectly reconstruct $z_x$ for any $(s, z_x) \sim P_{S,Z_X}$ under the constraint that $I(Z_X; S)$ is maximized, then $Z_X$ discards the task-irrelevant information, excluding $I(X; S|T)$. Similar to the forward predictive learning, we use $\mathbb{E}_{P_{S,Z_X}}[\log Q_\phi(Z_X|S)]$ as a lower bound of $\mathbb{E}_{P_{S,Z_X}}[\log P(Z_X|S)]$. In our deployment, we take the advantage of the design in equation 1 and let $Q_\phi(Z_X|S)$ be Gaussian $\mathcal{N}\left(Z_X|F_S(S), \sigma\mathbf{I}\right)$:

$$\mathrm{L}_{IP} := \max_{Z_S = F_S(S), Z_X = F_X(X)} \mathbb{E}_{z_s, z_x \sim P_{Z_S, Z_X}}\left[ -\|z_x - z_s\|_2^2 \right]. \tag{3}$$

Note that optimizing equation 3 alone results in a degenerated solution, e.g., learning $Z_X$ and $Z_S$ to be the same constant.

**Composing SSL Objectives (to extract task-relevant and discard task-irrelevant information simultaneously).** So far, we discussed how prior self-supervised learning approaches extract task-relevant information via the contrastive or the forward predictive learning objectives. Our analysis also inspires a new loss, the inverse predictive learning objective, to discard task-irrelevant information. Now, We present a composite loss to combine them together:

$$\mathrm{L}_{SSL} = \lambda_{CL}\mathrm{L}_{CL} + \lambda_{FP}\mathrm{L}_{FP} + \lambda_{IP}\mathrm{L}_{IP}, \tag{4}$$

where $\lambda_{CL}$, $\lambda_{FP}$, and $\lambda_{IP}$ are hyper-parameters. This composite loss enables us to extract task-relevant and discard task-irrelevant information simultaneously.

---

[5] $\mathbb{E}_{P_{S,Z_X}}[\log P(S|Z_X)] = \max_{Q_\phi} \mathbb{E}_{P_{S,Z_X}}[\log Q_\phi(S|Z_X)] + D_{\mathrm{KL}}\left(P(S|Z_X) \| Q_\phi(S|Z_X)\right) \geq \max_{Q_\phi} \mathbb{E}_{P_{S,Z_X}}[\log Q_\phi(S|Z_X)]$.

[6] The assumption of identity covariance in the Gaussian is only a particular parameterization of the distribution $Q(\cdot|\cdot)$. Other examples are MocoGAN (Tulyakov et al., 2018), which assumes $Q$ is Laplacian (i.e., $\ell_1$ reconstruction loss) and $\phi$ is a deconvolutional network (Long et al., 2015). Transformer-XL (Dai et al., 2019) assumes $Q$ is a categorical distribution (i.e., cross entropy loss) and $\phi$ is a Transformer network (Vaswani et al., 2017). Although Gaussian with diagonal covariance is not the best assumption, it is perhaps the simplest one.

## 2.4 Theoretical Analysis - Bayes Error Rate for Downstream Classification

In last subsection, we see the practical aspects of our designed SSL strategies. Now, we provide an theoretical analysis on the representations' generalization error when $T$ is a categorical variable . We use Bayes error rate as an example, which stands for the irreducible error (smallest generalization error (Feder & Merhav, 1994)) when learning an arbitrary classifier from the representation to infer the labels. In specific, let $P_e$ be the Bayes error rate of arbitrary learned representations $Z_X$ and $\hat{T}$ as the estimation for $T$ from our classifier, $P_e := \mathbb{E}_{z_x \sim P_{Z_X}}[1 - \max_{t \in T} P(\hat{T} = t | z_x)]$.

To begin with, we present a general form of sample complexity with mutual information ($I(Z_X; S)$) estimation using empirical samples from distribution $P_{Z_X,S}$. Let $P_{Z_X,S}^{(n)}$ denote the (uniformly sampled) empirical distribution of $P_{Z_X,S}$ and $\hat{I}_\theta^{(n)}(Z_X; S) := \mathbb{E}_{P_{Z_X;S}^{(n)}}[\hat{f}_\theta(z_x, s)]$ with $\hat{f}_\theta$ being the estimated log density ratio (i.e., $\log p(s|z_x)/p(s)$).

**Proposition 1** (Mutual Information Neural Estimation, restating Theorem 1 by Tsai et al. (2020)). *Let $0 < \delta < 1$. There exists $d \in \mathbb{N}$ and a family of neural networks $\mathcal{F} := \{\hat{f}_\theta : \theta \in \Theta \subseteq \mathbb{R}^d\}$ where $\Theta$ is compact, so that $\exists \theta^* \in \Theta$, with probability at least $1 - \delta$ over the draw of $\{z_{xi}, s_i\}_{i=1}^n \sim P_{Z_X,S}^{\otimes n}$,*

$$\left| \hat{I}_{\theta^*}^{(n)}(Z_X; S) - I(Z_X; S) \right| \leq O\left( \sqrt{\frac{d + \log(1/\delta)}{n}} \right).$$

This proposition shows that there exists a neural network $\theta^*$, with high probability, $\hat{I}_{\theta^*}^{(n)}(Z_X; S)$ can approximate $I(Z_X; S)$ with $n$ samples at rate $O(1/\sqrt{n})$. Under this network $\theta^*$ and the same parameters $d$ and $\delta$, we are ready to present our main results on the Bayes error rate. Formally, let $|T|$ be $T$'s cardinalitiy and $\mathrm{Th}(x) = \min\{\max\{x, 0\}, 1 - 1/|T|\}$ as a thresholding function:

**Theorem 3** (Bayes Error Rates for Arbitrary Learned Representations). *For an arbitrary learned representations $Z_X$, $P_e = \mathrm{Th}(\bar{P}_e)$ with*

$$\bar{P}_e \leq 1 - \exp\left( - \left( H(T) + I(X; S|T) + I(Z; X|S, T) - \hat{I}_{\theta^*}^{(n)}(Z_X; S) + O\left(\sqrt{\frac{d + \log(1/\delta)}{n}}\right) \right) \right).$$

Given arbitrary learned representations ($Z_X$), Theorem 3 suggests the corresponding Bayes error rate ($P_e$) is small when: 1) the estimated mutual information $\left(\hat{I}_{\theta^*}^{(n)}(Z_X; S)\right)$ is large; 2) a larger number of samples $n$ are used for estimating the mutual information; and 3) the task-irrelevant information $\big($the compression gap $I(X; S|T)$ and the superfluous information $I(Z; X|S, T)$, defined in Theorem 2$\big)$ is small. The first and the second results supports the claim that maximizing $I(Z_X; S)$ may learn the representations that are beneficial to downstream tasks. The third result implies the learned representations may perform better on the downstream task when the compression gap is small. Additionally, $Z^{\mathrm{sslmin}}$ is preferable than $Z^{\mathrm{ssl}}$ since $I(Z^{\mathrm{sslmin}}; X|S, T) = 0$ and $I(Z^{\mathrm{ssl}}; X|S, T) \geq 0$.

**Theorem 4** (Bayes Error Rates for Self-supervised Learned Representations). *Let $P_e^{\mathrm{sup}}/P_e^{\mathrm{ssl}}/P_e^{\mathrm{sslmin}}$ be the Bayes error rate of the supervised or the self-supervised learned representations $Z_X^{\mathrm{sup}}/Z_X^{\mathrm{ssl}}/Z_X^{\mathrm{sslmin}}$. Then, $P_e^{\mathrm{ssl}} = \mathrm{Th}(\bar{P}_e^{\mathrm{ssl}})$ and $P_e^{\mathrm{sslmin}} = \mathrm{Th}(\bar{P}_e^{\mathrm{sslmin}})$ with*

$$-\frac{\log(1 - P_e^{\mathrm{sup}}) + \log 2}{\log(|T|)} \leq \{\bar{P}_e^{\mathrm{ssl}}, \bar{P}_e^{\mathrm{sslmin}}\} \leq 1 - \exp\left( - (\log 2 + P_e^{\mathrm{sup}} \cdot \log|T| + \epsilon_{\mathrm{info}}) \right).$$

Given our self-supervised learned representations ($Z_X^{\mathrm{ssl}}$ and $Z_X^{\mathrm{sslmin}}$), Theorem 4 suggests a smaller upper bound of $P_e^{\mathrm{ssl}}$ (or $P_e^{\mathrm{sslmin}}$) when the redundancy between the input and the self-supervised signal ($\epsilon_{\mathrm{info}}$, defined in Assumption 1) is small. This result implies the self-supervised learned representations may perform better on the downstream task when the multi-view redundancy is small.

## 3 Controlled Experiments

This section aims at providing empirical supports for Theorems 1 and 2 and comparing different SSL objectives. In particular, we present information inequalities in Theorems 1 and 2 regarding the amount of the task-relevant and the task-irrelevant information that will be extracted and discarded when learning self-supervised representations. Nonetheless, quantifying the information is notoriously hard and often leads to inaccurate quantifications in practice (McAllester & Stratos, 2020; Song &

Figure 3: Comparisons for different compositions of SSL objectives on Omniglot and CIFAR10.

Ermon, 2019). Not to mention the information we aim to quantify is the conditional information, which is believed to be even more challenging than quantifying the unconditional one (Póczos & Schneider, 2012). To address this concern, we instead study the generalization error of the self-supervised learned representations, theoretically (Bayes error rate discussed in Section 2.4) and empirically (test performance discussed in this section).

Another important aspect of the experimental design is examining equation 4, which can be viewed as a Lagrangian relaxation to learn representations that contain minimal and sufficient self-supervision (see Definition 2): a weighted combination between $I(Z_X; S)$ and $-H(Z_X|S)$. In particular, the contrastive loss $L_{CL}$ and the forward-predictive loss $L_{FP}$ represent different realizations of modeling $I(Z_X; S)$ and the inverse-predictive loss $L_{FP}$ represents a realization of modeling $-H(Z_X|S)$.

We design two sets of experiments: The first one is when the input and self-supervised signals lie in the same modality (visual) and are likely to satisfy the multi-view redundancy assumption (Assumption 1). The second one is when the input and self-supervised signals lie in very different modalities (visual and textual), thus challenging the SSL objective's generalization ability.

**Experiment I - Visual Representation Learning.** We use Omniglot dataset (Lake et al., 2015) [7] in this experiment. The training set contains images from $964$ characters, and the test set contains $659$ characters. There are no characters overlap between the training and test set. Each character contains twenty examples drawn from twenty different people. We regard image as input ($X$) and generate self-supervised signal ($S$) by first sampling an image from the same character as the input image and then applying translation/ rotation to it. Furthermore, we represent task-relevant information ($T$) by the labels of the image. Under this self-supervised signal construction, the exclusive information in $X$ or $S$ are drawing styles (i.e., by different people) and image augmentations, and only their shared information contribute to $T$. To formally show the later, if $T$ representing the label for $X/S$, then $P(T|X)$ and $P(T|S)$ are Dirac. Hence, $T \perp\!\!\!\perp S|X$ and $T \perp\!\!\!\perp X|S$, suggesting Assumption 1 holds.

We train the feature mapping $F_X(\cdot)$ with SSL objectives (see eq. equation 4), set $F_S(\cdot) = F_X(\cdot)$, let $R(\cdot)$ be symmetrical to $F_X(\cdot)$, and $G(\cdot)$ be an identity mapping. On the test set, we fix the mapping and randomly select 5 examples per character as the labeled examples. Then, we classify the rest of the examples using the 1-nearest neighbor classifier based on feature (i.e., $Z_X = F_X(X)$) cosine similarity. The random performance on this task stands at $\frac{1}{659} \approx 0.15\%$. One may refer to Supplementary for more details.

▷ *Results & Discussions.* In Figure 3, we evaluate the generalization ability on the test set for different SSL objectives. First, we examine how the introduced inverse predictive learning objective $L_{IP}$ can help improve the performance along with the contrastive learning objective $L_{CL}$. We present the results in Figure 3 (a) and also provide experiments with SimCLR (Chen et al., 2020) on CIFAR10 (Krizhevsky et al., 2009) in Figure 3 (b), where $\lambda_{IP} = 0$ refers to the exact same setup as in SimCLR (which considers only $L_{CL}$). We find that adding $L_{IP}$ in the objective can boost model performance, although being sensitive to the hyper-parameter $\lambda_{IP}$. According to Theorem 2, the improved performance suggests a more compressed representation results in better performance for the downstream tasks. Second, we add the discussions with the forward predictive learning objective $L_{FP}$. We present the results in Figure 3 (c). Comparing to $L_{FP}$, $L_{CL}$ 1) reaches better test accuracy; 2) requires shorter training epochs to reach the best performance; and 3) suffers from overfitting with long-epoch training. Combining both of them ($L_{CL} + 0.005L_{FP}$) brings their advantages together.

**Experiment II - Visual-Textual Representation Learning.** We provide experiments using MS COCO dataset (Lin et al., 2014) that contains 328k multi-labeled images with 2.5 million labeled

---

[7]More complex datasets such as CIFAR10 (Krizhevsky et al., 2009) or ImageNet (Deng et al., 2009), to achieve similar performance, require a much larger training scale from contrastive to forward predictive objective. For example, on ImageNet, MoCo (He et al., 2019) uses 8 GPUs for its contrastive objective and ImageGPT (Chen et al.) uses 2048 TPUs for its forward predictive objective. We choose the Omniglot to ensure fair comparisons among different self-supervised learning objectives under reasonable computation constraint.

| (a) MS COCO (Using $L_{CL}$ as SSL objective) | | |
|---|---|---|
| Setting | Micro ROC-AUC | Subset Acc. |
| Cross-modality Self-supervised Learning | | |
| Raw BERT + Raw ResNet | $0.5963 \pm 0.0034$ | $0.0166 \pm 0.0017$ |
| Pre-trained BERT + Raw ResNet | $0.5915 \pm 0.0035$ | $0.0163 \pm 0.0011$ |
| Raw BERT + Pre-trained ResNet | $0.7049 \pm 0.0040$ | $0.2081 \pm 0.0063$ |
| Pre-trained BERT + Pre-trained ResNet | $0.7065 \pm 0.0026$ | $0.2123 \pm 0.0040$ |
| Non Self-supervised Learning | | |
| Only Pre-trained ResNet | $0.6761 \pm 0.0045$ | $0.1719 \pm 0.0015$ |

(b) Raw BERT + Pre-trained ResNet (Contrastive with Inverse Predictive)

Figure 4: Comparisons for different settings on self-supervised visual-textual representation training. We report metrics on MS COCO validation set with mean and standard deviation from 5 random trials.

instances from 91 objects. Each image has 5 annotated captions describing the relationships between objects in the scenes. We regard image as input ($X$) and its textual descriptions as self-supervised signal ($S$). Since vision and text are two very different modalities, the multi-view redundancy may not be satisfied, which means $\epsilon_{\text{info}}$ may be large in Assumption 1.

We adopt $L_{\text{CL}}$ ($+\lambda_{IP}L_{\text{IP}}$) as our SSL objective. We use ResNet18 (He et al., 2016) image encoder for $F_X(\cdot)$ (trained from scratch or fine-tuned on ImageNet (Deng et al., 2009) pre-trained weights), BERT-uncased (Devlin et al., 2018) text encoder for $F_S(\cdot)$ (trained from scratch or BookCorpus (Zhu et al., 2015)/Wikipedia pre-trained weights), and a linear layer for $G(\cdot)$. After performing self-supervised visual-textual representation learning, we consider the downstream multi-label classification over 91 categories. We evaluate learned visual representation ($Z_X$) using *downstream linear evaluation protocol* (Oord et al., 2018; Hénaff et al., 2019; Tian et al., 2019; Hjelm et al., 2018; Bachman et al., 2019; Tschannen et al., 2019). Specifically, a linear classifier is trained from the self-supervised learned (fixed) representation to the labels on the training set. Commonly used metrics for multi-label classification are reported on MS COCO validation set: Micro ROC-AUC and Subset Accuracy. One may refer to Supplementary for more details on these metrics.

▷ *Results & Discussions.* First, Figure 4 (a) suggests that the SSL strategy can still work when the input and self-supervised signals lie in different modalities. For example, pre-trained ResNet with BERT (either raw or the pre-trained one) outperforms pre-trained ResNet alone. We also see that the self-supervised learned representations benefit more if the ResNet is pre-trained but not the BERT. This result is in accord with the fact that object recognition requires more understanding in vision, and hence the pre-trained ResNet is preferrable than the pre-trained BERT. Next, Figure 4 (b) suggests that the self-supervised learned representations can be further improved by combining $L_{CL}$ and $L_{IP}$, suggesting $L_{IP}$ may be a useful objective to discard task-irrelevant information.

**Remarks on $\lambda_{IP}$ and $L_{IP}$.** As observed in the experimental results, $\lambda_{IP}$ is a sensitive hyper-parameter to the performance. We provide an optimization perspective to address this concern. Note that one of the our goals is to examine the setting when learning the minimal and sufficient representations for self-supervision (see Definition 2): minimize $H(Z_X|S)$ under the constraint that $I(Z_X; S)$ is maximized. However, this constrained optimization is not feasible when considering gradients methods in neural networks. Hence, our approach can be seen as its Lagrangian Relaxation by a weighted combination between $L_{CL}$ (or $L_{FP}$, representing $I(Z_X; S)$) and $L_{IP}$ (representing $H(Z_X|S)$) with the $\lambda_{IP}$ being the Lagrangian coefficient.

The optimal $\lambda_{IP}$ can be obtained by solving the Lagrangian dual, which depends on the parametrization of $L_{CL}$ (or $L_{FP}$) and $L_{IP}$. Different parameterizations lead to different loss and gradient landscapes, and hence the optimal $\lambda_{IP}$ differs across experiments. This conclusion is verified by the results presented in Figure 3 (a) and (b) and Figure 4 (b). Lastly, we point out that even not solving the Lagrangian dual, an empirical observation across experiments is that $\lambda_{IP}$ which leads to the best performance is when the scale of $L_{IP}$ is one-tenth to the scale of $L_{CL}$ (or $L_{FP}$).

## 4 RELATED WORK

Prior work by Arora et al. (2019) and the recent concurrent work (Lee et al., 2020; Tosh et al., 2020) are landmarks for theoretically understanding the success of SSL. In particular, Arora et al. (2019); Lee et al. (2020) showed a decreased sample complexity for downstream supervised tasks when adopting contrastive learning objectives (Arora et al., 2019) or predicting the known information in the data (Lee et al., 2020). Tosh et al. (2020) showed that the linear functions of the learned representations are nearly optimal on downstream prediction tasks. By viewing the input and the self-supervised signal as two corresponding views of the data, we discuss the differences among these works and ours. On the one hand, the work by Arora et al. (2019); Lee et al. (2020) assume strong independence between the views conditioning on the downstream tasks , i.e., $I(X; S|T) \approx 0$.

On the other hand, the work by Tosh et al. (2020) and ours assume strong independence between the downstream task and one view conditioning on the other view, i.e., $I(T; X|S) \approx 0$. Prior work (Balcan et al., 2005; Du et al., 2010) have compared these two assumptions and pointed out the former one ($I(X; S|T) \approx 0$) is too strong and not likely to hold in practice. We note that all these related work and ours have shown that the self-supervised learning methods are learning to extract task-relevant information. Our work additionally presents to discard task-irrelevant information and quantifies the amount of information that cannot be discarded.

Our method also resembles the InfoMax principle (Linsker, 1988; Hjelm et al., 2018) and the Multi-view Information Bottleneck method (Federici et al., 2020). The InfoMax principle aims at preserving the information of itself, while ours aims at extracting the information in the self-supervised signal. On the other hand, to reduce the redundant information across views, the Multi-view Information Bottleneck method proposed to minimize the conditional mutual information $I(Z_X; X|S)$, while ours propose to minimize the conditional entropy $H(Z_X|S)$. The conditional entropy minimization problem can be easily optimized via our proposed inversed predictive learning objective.

Another related work is InfoMin (Tian et al., 2020), where both InfoMin and our method suggest to learn the representations that contain "not" too much information. In particular, InfoMin presents to augment the data (i.e., by constructing learnable data augmentations) such that the shared information between augmented variants is as minimal as possible, followed by the mutual information maximization between the learned features from the augmented variants. Our method instead considers standard augmentations (e.g., rotations and translations), followed by learning representations that contain no more than the shared information between the augmented variants of the data.

On the empirical side, we explain why contrastive (Oord et al., 2018; Bachman et al., 2019; Chen et al., 2020) and predictive learning (Zhang et al., 2016; Pathak et al., 2016; Vondrick et al., 2016; Chen et al.) approaches can unsupervised extract task-relevant information. Different from these work, we present an objective to discard task-irrelevant information and show its combination with existing contrastive or predictive objectives benefits the performance.

## 5 CONCLUSION

This work studies both theoretical and empirical perspectives on self-supervised learning. We show that the self-supervised learned representations could extract task-relevant information (with a potential loss) and discard task-irrelevant information (with a fixed gap), along with their practical deployments such as contrastive and predictive learning objectives. We believe this work sheds light on the advantages of self-supervised learning and may help better understand when and why self-supervised learning is likely to work. In the future, we plan to connect our framework and recent SSL methods that cannot be easily fit into our analysis: e.g., BYOL (Grill et al., 2020), SWAV (Caron et al., 2020), and Unifromality-Alignment (Wang & Isola, 2020).

## ACKNOWLEDGEMENT

This work was supported in part by the NSF IIS1763562, NSF Awards #1750439 #1722822, National Institutes of Health, IARPA D17PC00340, ONR Grant N000141812861, and Facebook PhD Fellowship. We would also like to acknowledge NVIDIA's GPU support.

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

## A  REMARKS ON LEARNING MINIMAL AND SUFFICIENT REPRESENTATIONS

In the main text, we discussed the objectives to learn minimal and sufficient representations (Definition 1). Here, we discuss the similarities and differences between the prior methods (Tishby et al., 2000; Achille & Soatto, 2018) and ours. First, to obtain sufficient representations (for the downstream task $T$), all the methods presented to maximize $I(Z_X; T)$. Then, to maintain minimal amount of information in the representations, the prior methods (Tishby et al., 2000; Achille & Soatto, 2018) presented to minimize $I(Z_X; X)$ and the ours presents to minimize $H(Z_X|T)$. Our goal is to relate $I(Z_X; X)$ minimization and $H(Z_X|T)$ minimization in our framework.

To begin with, under the constraint $I(Z_X; T)$ is maximized, we see that minimizing $I(Z_X; X)$ is equivalent to minimizing $I(Z_X; X|T)$. The reason is that $I(Z_X; X) = I(Z_X; X|T) + I(Z_X; X; T)$, where $I(Z_X; X; T) = I(Z_X; T)$ due to the determinism from $X$ to $Z_X$ (our framework learns a deterministic function from $X$ to $Z_X$) and $I(Z_X; T)$ is maximized in our constraint. Then, $I(Z_X; X|T) = H(Z_X|T) - H(Z_X|X, T)$, where $H(Z_X|T)$ contains no randomness (no information) as $Z_X$ being deterministic from $X$. Hence, $I(Z_X; X|T)$ minimization and $H(Z_X|T)$ minimization are interchangeable.

The same claim can be made from the downstream task $T$ to the self-supervised signal $S$. In specific, when $X$ to $Z_X$ is deterministic, $I(Z_X; X|S)$ minimization and $H(Z_X|S)$ minimization are interchangeable. As discussed in the related work section, for reducing the amount of the redundant information, Federici et al. (2020) presented to use $I(Z_X; X|S)$ minimization and ours presented to use $H(Z_X|T)$ minimization. We also note that directly minimizing the conditional mutual information (i.e., $I(Z_X; X|S)$) requires a min-max optimization (Mukherjee et al., 2020), which may cause instability in practice. To overcome the issue, Federici et al. (2020) assumes a Gaussian encoder for $X \to Z_X$ and presents an upper bound of the original objective.

## B  PROOFS FOR THEOREM 1 AND 2

We start by presenting a useful lemma from the fact that $F_X(\cdot)$ is a deterministic function:

**Lemma 1** (Determinism). *If $P(Z_X|X)$ is Dirac, then the following conditional independence holds: $T \perp\!\!\!\perp Z_X|X$ and $S \perp\!\!\!\perp Z_X|X$, inducing a Markov chain $S \leftrightarrow T \leftrightarrow X \to Z_X$.*

*Proof.* When $Z_X$ is a deterministic function of $X$, for any $A$ in the sigma-algebra induced by $Z_X$ we have $\mathbb{E}[\mathbf{1}_{[Z_X \in A]}|X, \{T, S\}] = \mathbb{E}[\mathbf{1}_{[Z_X \in A]}|X, S] = \mathbb{E}[\mathbf{1}_{[Z_X \in A]}|X]$, which implies $T \perp\!\!\!\perp Z_X|X$ and $S \perp\!\!\!\perp Z_X|X$. □

Theorem 1 and 2 in the main text restated:

**Theorem 5** (Task-relevant information with a potential loss $\epsilon_{\text{info}}$, restating Theorem 1 in the main text). *The supervised learned representations (i.e., $I(Z_X^{\text{sup}}; T)$ and $I(Z_X^{\text{sup}_{\min}}; T)$) contain all the task-relevant information in the input (i.e., $I(X; T)$). The self-supervised learned representations (i.e., $I(Z_X^{\text{ssl}}; T)$ and $I(Z_X^{\text{ssl}_{\min}}; T)$) contain all the task-relevant information in the input with a potential loss $\epsilon_{\text{info}}$. Formally,*

$$I(X; T) = I(Z_X^{\text{sup}}; T) = I(Z_X^{\text{sup}_{\min}}; T) \geq I(Z_X^{\text{ssl}}; T) \geq I(Z_X^{\text{ssl}_{\min}}; T) \geq I(X; T) - \epsilon_{\text{info}}.$$

*Proof.* The proofs contain two parts. The first one is showing the results for the supervised learned representations and the second one is for the self-supervised learned representations.

*Supervised Learned Representations:* Adopting Data Processing Inequality (DPI by Cover & Thomas (2012)) in the Markov chain $S \leftrightarrow T \leftrightarrow X \rightarrow Z_X$ (Lemma 1), $I(Z_X; T)$ is maximized at $I(X; T)$. Since both supervised learned representations ($Z_X^{\text{sup}}$ and $Z_X^{\text{sup}_{\min}}$) maximize $I(Z_X; T)$, we conclude $I(Z_X^{\text{sup}}; T) = I(Z_X^{\text{sup}_{\min}}; T) = I(X; T)$.

*Self-supervised Learned Representations:* First, we have

$$I(Z_X; S) = I(Z_X; T) - I(Z_X; T|S) + I(Z_X; S|T) = I(Z_X; T; S) + I(Z_X; S|T)$$

and

$$I(X; S) = I(X; T) - I(X; T|S) + I(X; S|T) = I(X; T; S) + I(X; S|T).$$

By DPI in the Markov chain $S \leftrightarrow T \leftrightarrow X \rightarrow Z_X$ (Lemma 1), we know

- $I(Z_X; S)$ is maximized at $I(X; S)$

- $I(Z_X; S; T)$ is maximized at $I(X; S; T)$

- $I(Z_X; S|T)$ is maximized at $I(X; S|T)$

Since both self-supervised learned representations ($Z_X^{\text{ssl}}$ and $Z_X^{\text{ssl}_{\min}}$) maximize $I(Z_X; S)$, we have $I(Z_X^{\text{ssl}}; S) = I(Z_X^{\text{ssl}_{\min}}; S) = I(X; S)$. Hence, $I(Z_X^{\text{ssl}}; S; T) = I(Z_X^{\text{ssl}_{\min}}; S; T) = I(X; S; T)$ and $I(Z_X^{\text{ssl}}; S|T) = I(Z_X^{\text{ssl}_{\min}}; S|T) = I(X; S|T)$. Using the result $I(Z_X^{\text{ssl}}; S; T) = I(Z_X^{\text{ssl}_{\min}}; S; T) = I(X; S; T)$, we get

$$I(Z_X^{\text{ssl}}; T) = I(X; T) - I(X; T|S) + I(Z_X^{\text{ssl}}; T|S)$$

and

$$I(Z_X^{\text{ssl}_{\min}}; T) = I(X; T) - I(X; T|S) + I(Z_X^{\text{ssl}_{\min}}; T|S).$$

Now, we are ready to present the inequalities:

1. $I(X; T) \geq I(Z_X^{\text{ssl}}; T)$ due to $I(X; T|S) \geq I(Z_X^{\text{ssl}}; T|S)$ by DPI.

2. $I(Z_X^{\text{ssl}}; T) \geq I(Z_X^{\text{ssl}_{\min}}; T)$ due to $I(Z_X^{\text{ssl}}; T|S) \geq I(Z_X^{\text{ssl}_{\min}}; T|S) = 0$. Since $H(Z_X|S)$ is minimized at $Z_X^{\text{ssl}_{\min}}$, $I(Z_X^{\text{ssl}_{\min}}; T|S) = 0$.

3. $I(Z_X^{\text{ssl}_{\min}}; T) \geq I(X; T) - \epsilon_{\text{info}}$ due to

   $$I(X; T) - I(X; T|S) + I(Z_X^{\text{ssl}_{\min}}; T|S) \geq I(X; T) - I(X; T|S) \geq I(X; T) - \epsilon_{\text{info}},$$

   where $I(X; T|S) \leq \epsilon_{\text{info}}$ by the redundancy assumption.

$\square$

**Theorem 6** (Task-irrelevant information with a fixed compression gap $I(X; S|T)$, restating Theorem 2 in the main text). *The sufficient self-supervised representation (i.e., $I(Z_X^{\text{ssl}}; T)$) contains more task-irrelevant information in the input than then the sufficient and minimal self-supervised representation (i.e., $I(Z_X^{\text{ssl}_{\min}}; T)$). The later contains an amount of the information, $I(X; S|T)$, that cannot be discarded from the input. Formally,*

$$I(Z_X^{\text{ssl}}; X|T) = I(X; S|T) + I(Z_X^{\text{ssl}}; X|S, T) \geq I(Z_X^{\text{ssl}_{\min}}; X|T) = I(X; S|T) \geq I(Z_X^{\text{sup}_{\min}}; X|T) = 0.$$

*Proof.* First, we see that

$$I(Z_X; X|T) = I(Z_X; X; S|T) + I(Z_X; X|S, T) = I(Z_X; S|T) + I(Z_X; X|S, T),$$

where $I(Z_X; X; S|T) = I(Z_X; S|T)$ by DPI in the Markov chain $S \leftrightarrow T \leftrightarrow X \to Z_X$.

We conclude the proof by combining the following:

- From the proof in Theorem 5, we showed $I(Z_X^{\text{ssl}}; S|T) = I(Z_X^{\text{ssl}_{\min}}; S|T) = I(X; S|T)$.

- Since $H(Z_X|S)$ is minimized at $Z_X^{\text{ssl}_{\min}}$, $I(Z_X^{\text{ssl}_{\min}}; X|S, T) = 0$.

- Since $H(Z_X|T)$ is minimized at $Z_X^{\text{sup}_{\min}}$, $I(Z_X^{\text{sup}_{\min}}; X|T) = 0$.

$\square$

## C    PROOF FOR PROPOSITION 1

**Proposition 2** (Mutual Information Neural Estimation, restating Proposition 1 in the main text). *Let $0 < \delta < 1$. There exists $d \in \mathbb{N}$ and a family of neural networks $\mathcal{F} := \{\hat{f}_\theta : \theta \in \Theta \subseteq \mathbb{R}^d\}$ where $\Theta$ is compact, so that $\exists \theta^* \in \Theta$, with probability at least $1 - \delta$ over the draw of $\{z_{x_i}, s_i\}_{i=1}^n \sim P_{Z_X, S}^{\otimes n}$,*

$$\left| \widehat{I}_{\theta^*}^{(n)}(Z_X; S) - I(Z_X; S) \right| \le O\left( \sqrt{\frac{d + \log(1/\delta)}{n}} \right).$$

*Sketch of Proof.* The proof is a standard instance of uniform convergence bound. First, we assume the boundness and the Lipschitzness of $\hat{f}_\theta$. Then, we use the universal approximation lemma of neural networks (Hornik et al.). Last, combing all these two along with the uniform convergence in terms of the covering number (Bartlett, 1998), we complete the proof.    $\square$

We note that the complete proof can be found in the prior work (Tsai et al., 2020). An alternative but similar proof can be found in another prior work (Belghazi et al., 2018), which gives us $\left| \widehat{I}_{\theta^*}^{(n)}(Z_X; S) - I(Z_X; S) \right| \le O\left( \sqrt{\frac{d\log d + \log(1/\delta)}{n}} \right)$. The subtle difference between them is that, given a neural network function space $\Theta \subseteq \mathbb{R}^d$ and its covering number $\mathcal{N}(\Theta, \eta)$, Tsai et al. (2020) has $\mathcal{N}(\Theta, \eta) = O\left( (\eta)^{-d} \right)$ by Bartlett (1998) and Belghazi et al. (2018) has $\mathcal{N}(\Theta, \eta) = O\left( (\eta/\sqrt{d})^{-d} \right)$ by Shalev-Shwartz & Ben-David (2014). Both are valid and the one used by Tsai et al. (2020) is tighter.

## D    PROOFS FOR THEOREM 3 AND 4

To begin with, we see that

$$\begin{aligned}
I(Z_X; T) &= I(Z_X; X) - I(Z_X; X|T) + I(Z_X; T|X) = I(Z_X; X) - I(Z_X; X|T) \\
&= I(Z_X; S) - I(Z_X; S|X) + I(Z_X; X|S) - I(Z_X; X|T) \\
&= I(Z_X; S) + I(Z_X; X|S) - I(Z_X; X|T) \\
&\ge I(Z_X; S) - I(Z_X; X|T),
\end{aligned}$$

where $I(Z_X; T|X) = I(Z_X; S|X) = 0$ due to the determinism from $X$ to $Z_X$. Then, in the proof of Theorem 6, we have shown $I(Z_X; X|T) = I(Z_X; S|T) + I(Z_X; X|S, T)$. Hence,

$$\begin{aligned}
I(Z_X; T) &\ge I(Z_X; S) - I(Z_X; S|T) - I(Z_X; X|S, T) \\
&\ge I(Z_X; S) - I(X; S|T) - I(Z_X; X|S, T),
\end{aligned}$$

where $I(Z_X; S|T) \le I(X; S|T)$ by DPI.

Theorem 3 and 4 in the main text restated:

**Theorem 7** (Bayes Error Rates for Arbitrary Learned Representations, restating Theorem 3 in the main text). *For an arbitrary learned representations $Z_X$, $P_e = \text{Th}(\bar{P}_e)$ with*

$$\bar{P}_e \leq 1 - \exp^{-\left(H(T)+I(X;S|T)+I(Z;X|S,T)-\hat{I}_{\theta^*}^{(n)}(Z_X;S)+O\left(\sqrt{\frac{d+\log(1/\delta)}{n}}\right)\right)}.$$

*Proof.* We use the inequality between $P_e$ and $H(T|Z_X)$ indicated by Feder & Merhav (1994):

$$-\log(1 - P_e) \leq H(T|Z_X).$$

Combining with $I(Z_X;T) = H(T) - H(T|Z_X)$ and $I(Z_X;T) \geq I(Z_X;S) - I(X;S|T) - I(Z_X;X|S,T)$, we have

$$\log(1 - P_e) \geq -H(T) + I(Z_X;S) - I(X;S|T) - I(Z_X;X|S,T).$$

Hence,

$$P_e \leq 1 - \exp^{-\left(H(T)+I(X;S|T)+I(Z;X|S,T)-I(Z_X;S)\right)}.$$

Next, by definition of the Bayes error rate, we know $0 \leq P_e \leq 1 - \frac{1}{|T|}$.

We conclude the proof by combining Proposition 2, $\left|\hat{I}_{\theta^*}^{(n)}(Z_X;S) - I(Z_X;S)\right| \leq O\left(\sqrt{\frac{d+\log(1/\delta)}{n}}\right)$. $\quad\square$

**Theorem 8** (Bayes Error Rates for Self-supervised Learned Representations, restating Theorem 4 in the main text). *Let $P_e^{\text{sup}}/P_e^{\text{ssl}}/P_e^{\text{ssl}_{\min}}$ be the Bayes error rate of the supervised or the self-supervised learned representations $Z_X^{\text{sup}}/Z_X^{\text{ssl}}/Z_X^{\text{ssl}_{\min}}$. Then, $P_e^{\text{ssl}} = \text{Th}(\bar{P}_e^{\text{ssl}})$ and $P_e^{\text{ssl}_{\min}} = \text{Th}(\bar{P}_e^{\text{ssl}_{\min}})$ with*

$$-\frac{\log\left(1 - P_e^{\text{sup}}\right) + \log 2}{\log\left(|T|\right)} \leq \{\bar{P}_e^{\text{ssl}}, \bar{P}_e^{\text{ssl}_{\min}}\} \leq 1 - \exp^{-(\log 2 + P_e^{\text{sup}}\cdot\log|T|+\epsilon_{\text{info}})}.$$

*Proof.* We use the two inequalities between $P_e$ and $H(T|Z_X)$ by Feder & Merhav (1994) and Cover & Thomas (2012):

$$-\log(1 - P_e) \leq H(T|Z_X)$$

and

$$H(T|Z_X) \leq \log 2 + P_e\log|T|.$$

Combining the results from Theorem 5:

$$I(Z_X^{\text{sup}};T) \geq I(Z_X^{\text{ssl}};T) \geq I(Z_X^{\text{ssl}_{\min}};T) \geq I(Z_X^{\text{sup}};T) - \epsilon_{\text{info}},$$

we have

- the upper bound of the self-supervised learned representations' Bayes error rate:

$$\{-\log(1 - P_e^{\text{ssl}}), -\log(1 - P_e^{\text{ssl}_{\min}})\} \leq \{H(T|Z_X^{\text{ssl}}), H(T|Z_X^{\text{ssl}_{\min}})\}$$
$$\leq H(T|Z_X^{\text{sup}}) + \epsilon_{\text{info}}$$
$$\leq \log 2 + P_e^{\text{sup}}\log|T| + \epsilon_{\text{info}},$$

  which suggests $\{P_e^{\text{ssl}}, P_e^{\text{ssl}_{\min}}\} \leq 1 - \exp^{-(\log 2 + P_e^{\text{sup}}\cdot\log|T|+\epsilon_{\text{info}})}$.

- the lower bound of the self-supervised learned representations' Bayes error rate:

$$-\log(1 - P_e^{\text{sup}}) \leq H(T|Z_X^{\text{sup}})$$
$$\leq \{H(T|Z_X^{\text{ssl}}), H(T|Z_X^{\text{ssl}_{\min}})\}$$
$$\leq \{\log 2 + P_e^{\text{ssl}}\log|T|, \leq \{\log 2 + P_e^{\text{ssl}_{\min}}\log|T|\},$$

  which suggests $-\frac{\log\left(1 - P_e^{\text{sup}}\right)+\log 2}{\log\left(|T|\right)} \leq \{P_e^{\text{ssl}}, P_e^{\text{ssl}_{\min}}\}$.

We conclude the proof by having $P_e$ lie in the feasible range: $0 \leq P_e \leq 1 - \frac{1}{|T|}$. $\quad\square$

# E  TIGHTER BOUNDS FOR THE BAYES ERROR RATES

We note that the bound used in Theorems 7 and 8: $-\log(1 - P_e) \leq H(T|Z_X) \leq \log 2 + P_e \log|T|$ is not tight. A tighter bound is $H^-(P_e) \leq H(T|Z_X) \leq H^+(P_e)$ with

$$H^-(P_e) := H\Big(k(1 - P_e)\Big) + k(1 - P_e)\log k \text{ when } \frac{k-1}{k} \leq P_e \leq \frac{k}{k+1} \,,\, 1 \leq k \leq |T| - 1,$$

and
$$H^+(P_e) := H(P_e) + P_e \log(|T| - 1),$$

where $H(x) = -x\log(x) - (1 - x)\log(1 - x)$.

It is clear that $-\log(1 - P_e) \leq H^-(P_e)$ and $H^+(P_e) \leq \log 2 + P_e \log(|T|)$.

Hence, Theorem 7 and 8 can be improved as follows:

**Theorem 9** (Tighter Bayes Error Rates for Arbitrary Learned Representations). *For an arbitrary learned representations $Z_X$, $P_e = \mathrm{Th}(\bar{P}_e)$ with $\bar{P}_e \leq P_{e\,\mathrm{upper}}$. $P_{e\,\mathrm{upper}}$ is derived from the program*

$$\arg\max_{P_e} H^-(P_e) \leq H(T) - \hat{I}_\theta^{(n)}(Z_X^{\mathrm{ssl}}; S) + I(X; S|T) + I(Z_X; X|S, T) + O\Big(\sqrt{\frac{d + \log(1/\delta)}{n}}\Big).$$

**Theorem 10** (Tighter Bayes Error Rates for Self-supervised Learned Representations). *Let $P_e^{\mathrm{sup}}/P_e^{\mathrm{ssl}}/P_e^{\mathrm{ssl_{min}}}$ be the Bayes error rate of the supervised or the self-supervised learned representations $Z_X^{\mathrm{sup}}/Z_X^{\mathrm{ssl}}/Z_X^{\mathrm{ssl_{min}}}$. Then, $P_e^{\mathrm{ssl}} = \mathrm{Th}(\bar{P}_e^{\mathrm{ssl}})$ and $P_e^{\mathrm{ssl_{min}}} = \mathrm{Th}(\bar{P}_e^{\mathrm{ssl_{min}}})$ with*

$$P_{e\,\mathrm{lower}}^{\mathrm{ssl}} \leq \{\bar{P}_e^{\mathrm{ssl}}, \bar{P}_e^{\mathrm{ssl_{min}}}\} \leq P_{e\,\mathrm{upper}}^{\mathrm{ssl}}.$$

$P_{e\,\mathrm{lower}}^{\mathrm{ssl}}$ *is derived from the following program*

$$\arg\min_{P_e^{\mathrm{ssl}}} H^-(P_e^{\mathrm{sup}}) \leq H^+(P_e^{\mathrm{ssl}})$$

*and* $P_{e\,\mathrm{upper}}^{\mathrm{ssl}}$ *is derived from the following program*

$$\arg\max_{P_e^{\mathrm{ssl}}} H^-(P_e^{\mathrm{ssl}}) \leq H^+(P_e^{\mathrm{sup}}) + \epsilon_{\mathrm{info}}.$$

# F  MORE ON VISUAL REPRESENTATION LEARNING EXPERIMENTS

In the main text, we design controlled experiments on self-supervised visual representation learning to empirically support our theorem and examine different compositions of SSL objectives. In this section, we will discuss 1) the architecture design; 2) different deployments of contrastive/ forward predictive learning; and 3) different self-supervised signal construction strategy. We argue that these three additional set of experiments may be interesting future work.

## F.1  ARCHITECTURE DESIGN

The input image has size $105 \times 105$. For image augmentations, we adopt 1) rotation with degrees from $-10°$ to $+10°$; 2) translation from $-15$ pixels to $+15$ pixels; 3) scaling both width and height from $0.85$ to $1.0$; 4) scaling width from $0.85$ to $1.25$ while fixing the height; and 5) resizing the image to $28 \times 28$. Then, a deep network takes a $28 \times 28$ image and outputs a $1024-$dim. feature vector. The deep network has the structure: $\mathrm{Conv} - \mathrm{BN} - \mathrm{ReLU} - \mathrm{Conv} - \mathrm{BN} - \mathrm{ReLU} - \mathrm{MaxPool} - \mathrm{Conv} - \mathrm{BN} - \mathrm{ReLU} - \mathrm{MaxPool} - \mathrm{Conv} - \mathrm{BN} - \mathrm{ReLU} - \mathrm{MaxPool} - \mathrm{Flatten} - \mathrm{Linear} - \mathrm{L2Norm}$. Conv has 3x3 kernel size with 128 output channels, MaxPool has 2x2 kernel size, and Linear is a 1152 to 1024 weight matrix. $R(\cdot)$ is symmetric to $F_X(\cdot)$, which has $\mathrm{Linear} - \mathrm{BN} - \mathrm{ReLU} - \mathrm{UnFlatten} - \mathrm{DeConv} - \mathrm{BN} - \mathrm{ReLU} - \mathrm{DeConv} - \mathrm{BN} - \mathrm{ReLU} - \mathrm{DeConv} - \mathrm{BN} - \mathrm{ReLU} - \mathrm{DeConv}$. $R(\cdot)$ has the exact same number of parameters as $F_X(\cdot)$. Note that we use the same network designs in $I(\cdot, \cdot)$ and $H(\cdot|\cdot)$ estimations. To reproduce the results in our experimental section, please refer to our released code[8].

---

[8] https://github.com/yaohungt/Self_Supervised_Learning_Multiview

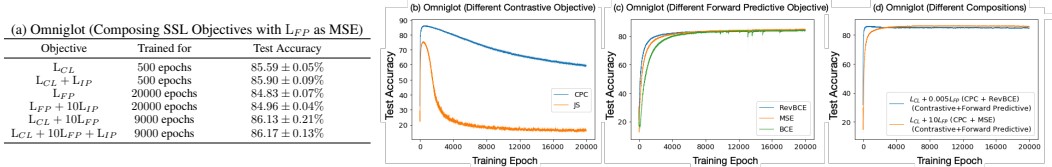

| (a) Omniglot (Composing SSL Objectives with $L_{FP}$ as MSE) | | |
|---|---|---|
| Objective | Trained for | Test Accuracy |
| $L_{CL}$ | 500 epochs | $85.59 \pm 0.05\%$ |
| $L_{CL} + L_{IP}$ | 500 epochs | $85.90 \pm 0.09\%$ |
| $L_{FP}$ | 20000 epochs | $84.83 \pm 0.07\%$ |
| $L_{FP} + 10L_{IP}$ | 20000 epochs | $84.96 \pm 0.04\%$ |
| $L_{CL} + 10L_{FP}$ | 9000 epochs | $86.13 \pm 0.21\%$ |
| $L_{CL} + 10L_{FP} + L_{IP}$ | 9000 epochs | $86.17 \pm 0.13\%$ |

Figure 5: Comparisons for different objectives/compositions of SSL objectives on self-supervised visual representation training. We report mean and its standard error from 5 random trials.

### F.2 DIFFERENT DEPLOYMENTS FOR CONTRASTIVE AND PREDICTIVE LEARNING OBJECTIVES

In the main text, for practical deployments, we suggest Contrastive Predictive Coding (CPC) Oord et al. (2018) for $L_{CL}$ and assume Gaussian distribution for the variational distributions in $L_{FP}$/ $L_{IP}$. The practical deployments can be abundant by using different mutual information approximations for $L_{CL}$ and having different distribution assumptions for $L_{FP}$/ $L_{IP}$. In the following, we discuss a few examples.

**Contrastive Learning.** Other than CPC Oord et al. (2018), another popular contrastive learning objective is JS Bachman et al. (2019), which is the lower bound of Jensen-Shannon divergence between $P(Z_S, Z_X)$ and $P(Z_S)P(Z_X)$ (a variational bound of mutual information). Its objective can be written as

$$\max_{Z_S = F_S(S), Z_X = F_X(X), G} \mathbb{E}_{P(Z_S, Z_X)}\big[-\text{softplus}(-\langle G(z_x), G(z_s)\rangle)\big] - \mathbb{E}_{P(Z_S)P(Z_X)}\big[\text{softplus}(\langle G(z_x), G(z_s)\rangle)\big],$$

where we use softplus to denote $\text{softplus}(x) = \log(1 + \exp(x))$.

**Predictive Learning.** Gaussian distribution may be the simplest distribution form that we can imagine, which leads to Mean Square Error (MSE) reconstruction loss. Here, we use forward predictive learning as an example, and we discuss the case when $\mathcal{S}$ lies in discrete $\{0, 1\}$ sample space. Specifically, we let $Q_\phi(S|Z_X)$ be factorized multivariate Bernoulli:

$$\max_{Z_X = F_X(X), R} \mathbb{E}_{P_S, Z_X} \left[ \sum_{i=1}^{p} s_i \cdot \log\left[R(z_x)\right]_i + (1 - s_i) \cdot \log\left[1 - R(z_x)\right]_i \right]. \tag{5}$$

This objective leads to Binary Cross Entropy (BCE) reconstruction loss.

If we assume each reconstruction loss corresponds to a particular distribution form, then by ignoring which variatioinal distribution we choose, we are free to choose arbitrary reconstruction loss. For instance, by switching $s$ and $z$ in eq. equation 5, the objective can be regarded as Reverse Binary Cross Entropy Loss (RevBCE) reconstruction loss. In our experiments, we find RevBCE works the best among {MSE, BCE, and RevBCE}. Therefore, in the main text, we choose RevBCE as the example reconstruction loss as $L_{FP}$.

**More Experiments.** We provide an additional set of experiments by having {CPC, JS} for $L_{CL}$ and {MSE, BCE, RevBCE} reconstruction loss for $L_{FP}$ in Figure 5. From the results, we find different formulation of objectives bring very different test generalization performance. We argue that, given a particular task, it is challenging but important to find the best deployments for contrastive and predictive learning objectives.

### F.3 DIFFERENT SELF-SUPERVISED SIGNAL CONSTRUCTION STRATEGY

In the main text, we design a self-supervised signal construction strategy that the input $(X)$ and the self-supervised signal $(S)$ differ in {drawing styles, image augmentations}. This self-supervised signal construction strategy is different from the one that is commonly adopted in most self-supervised visual representation learning work Tian et al. (2019); Bachman et al. (2019); Chen et al. (2020). Specifically, prior work consider the difference between input and the self-supervised signal only in image augmentations. We provide additional experiments in Fig. 6 to compare these two different self-supervised signal construction strategies.

We see that, comparing to the common self-supervised signal construction strategy Tian et al. (2019); Bachman et al. (2019); Chen et al. (2020), the strategy introduced in our controlled experiments has much better generalization ability to test set. It is worth noting that, although our construction

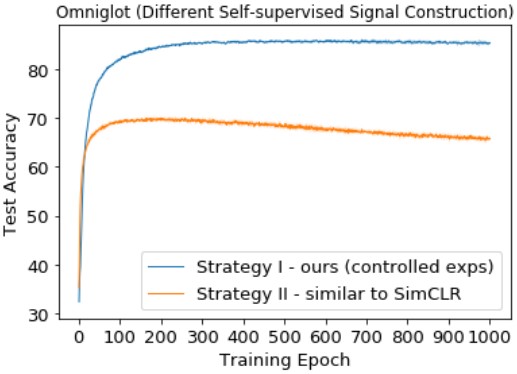

Figure 6: Comparisons for different self-supervised signal construction strategies. The differences between the input and the self-supervised signals are {drawing styles, image augmentations} for our construction strategy and only {image augmentations} for SimCLR Chen et al. (2020)'s strategy. We choose $L_{CL}$ as our objective, reporting mean and its standard error from 5 random trials.

strategy has access to the label information (i.e., we sample the self-supervised signal image from the same character with the input image), our SSL objectives do not train with the labels. Nonetheless, since we implicitly utilize the label information in our self-supervised construction strategy, it will be unfair to directly compare our strategy and prior one. An interesting future research direction is examining different self-supervised signal construction strategy and even combine full/part of label information into self-supervised learning.

## G  METRICS IN VISUAL-TEXTUAL REPRESENTATION LEARNING

- Subset Accuracy ($A$) Sorower, also know as the Exact Match Ratio (MR), ignores all partially correct (consider them incorrect) outputs and extend accuracy from the single label case to the multi-label setting.

$$MR = \frac{1}{n} \sum_{i=1}^{n} \mathbb{1}_{[Y_i = H_i]}$$

- Micro AUC ROC score Fawcett (2006) computes the AUC (Area under the curve) of a receiver operating characteristic (ROC) curve.

