# OpenReview forum: "Self-supervised Learning from a Multi-view Perspective"
_ICLR.cc/2021/Conference — ICLR 2021 Poster_

### Official Review · AnonReviewer4 · 2020-10-28
**This work presents a very nice theoretical analysis of self-supervised loss formulations but lacks solid experimental results**

**Rating:** 6
**Confidence:** 3

**Review:**

In this work the authors focus on self-supervised representation learning. This work proposes a composite loss function which includes contrastive loss, predictive loss, and a novel inverse predictive loss. The authors have provided theoretical analysis as well as experimental results on multiple datasets under control environment.

Pros:

This work presents a very detailed theoretical analysis for self-supervised learning objectives.

The idea of inverse predictive learning for filtering task irrelevant information is interesting.

Cons:

The inverse predictive learning appears to be similar to contrastive learning if we ignore the negative samples in the contrastive learning formulation and use the projection head from contrastive learning for inverse predictive learning. How will the authors differentiate between these two formulations?

The variation in the performance shown in Figure 3 is very marginal. Also, only selective results are shown in Figure 3. No results are shown on CIFAR-10 which uses all the three losses at the same time.  Figure 5 a shows some results on Omniglot, but the improvement shown there is very marginal. Also, how was the loss weights determined? Experiment on a large scale dataset, maybe ImageNet will be very useful to demonstrate the effectiveness of the proposed loss.

The weights required for inverse predictive learning in the loss formulation is not trivial. As shown in Figure 3, it varies with different datasets and varies a lot (1 for Omniglot and 0.1 for CIFAR10). Is there a simple way to determine this weights without exhaustive search on target dataset?

This work mainly focuses on the theoretical aspect of semi-supervised learning and have also presented an interesting inverse predictive learning task for self-supervision which can dis-regard task irrelevant information. However, it is not clear from the experimental results if this is really effective. The results shown in Figure 3 and Figure 4 are not sufficient to demonstrate its effectiveness. Also, quantitative comparison with existing approaches will also strengthen this work, which is only partially addressed. A more in-depth experimental analysis (apart from the theoretical analysis) is required to show its usefulness.

---

> ### Author Response · Authors · 2020-11-16
> **Response**
>
> [Remarks on the formulations of inverse predictive learning and contrastive learning]
>
> We are happy to talk about when the formulations between the inverse predictive learning and contrastive learning are similar.
>
> On the one hand, if we choose to deploy the inverse predictive learning loss to be a variational lower bound of -H(Z_X|S) with the variational distribution to be Gaussian, the loss considers an l2-difference between Z_X and Z_S. Nonetheless, if the variational distribution is Laplacian, the loss considers the l1-difference between Z_X and Z_S.
>
> On the other hand, if choosing the contrastive learning loss to be the Normalized Temperature-scaled Cross Entropy loss [1], then the loss term on the positive pair is the cosine similarity between the l2-normalized Z_X and Z_S. Then,
> cosine(Z_X, Z_S) = ( ||Z_X||_2^2 + ||Z_S||_2^2 - ||Z_X-Z_S||_2^2 ) /2 = 1 -  ||Z_X-Z_S||_2^2 / 2.
> Hence, when Z_X and Z_S are l2-normalized, the cosine similarity is equivalent to a scale and constant shift from the l2-difference. Nonetheless, if the features are not l2-normalized, the equivalence between the cosine similarity and the l2-difference no longer holds.
>
> As a summary, the formulations between the inverse predictive learning loss and the contrastive learning loss share similarities at specific choices on the loss function designs.
>
>
> [Remarks on the composite objective and the role of λ_IP]
>
> We are happy to provide more discussions on the objective design and the role of λ_IP in the composite objective. In particular, we provide an optimization perspective to motivate the design and we will include the discussion in the revised manuscript and highlight it in red.
>
> The design of the composite objective is motivated by learning minimal and sufficient representations for self-supervision (see Definition 2 in the paper): minimize H(Z_X|S) under the constraint that I(Z_X; S) is maximized. However, this constrained optimization is not feasible when considering gradients methods in neural networks. Hence, we consider its practical Lagrangian Relaxation by a weighted combination between L_CL (or L_FP, representing I(Z_X; S)) and L_IP (representing H(Z_X|S)) with the λ_IP being the Lagrangian coefficient. In summary, the linear combination of three tasks is a practical relaxation of the original constrained optimization problem.
>
> Then, we discuss the choice of λ_IP. The optimal λ_IP can be obtained by solving the Lagrangian dual, which depends on the parametrization of L_CL (or L_FP) and L_IP. Different parameterizations lead to different loss and gradient landscapes, and hence the optimal λ_IP differs across experiments and datasets. This conclusion is verified by the results presented in Figure 3(a), 3(b) and 4(b). Lastly, we point out that even not solving the Lagrangian dual, an empirical observation across experiments is that λ_IP which leads to the best performance is when the scale of L_IP is one-tenth to the scale of L_CL (or L_FP).
>
>
>
> [1] Chen et al., “A Simple Framework for Contrastive Learning of Visual Representations”, ICML 2020.

---

### Official Review · AnonReviewer1 · 2020-10-28
**An information-theoretical point to better understand SSL from a multi-view perspective, but the experimental part needs to be enhanced**

**Rating:** 6
**Confidence:** 4

**Review:**

This paper explores both theoretical and empirical perspectives on self-supervised learning. The authors attempt to prove that self-supervised learning could extract task-relevant information while discard task-irrelevant information. They also provide a composite objective that includes contrastive and predictive loss functions along with an additional regularization. Controlled experiments are conducted to support the presented theorems.

There are three major concerns for me.

1)	The experiments are conducted in a controlled way, including visual representation learning and visual-textual representation learning. Traditional uncontrolled experiments, such as unsupervised learning on Cifar10 or ImageNet are suggested.

2)	The results in Fig. 3 and Fig. 5 demonstrate that the performance is sensitive to the hyper-parameter of $L_{IP}$. How to set the hyper-parameter $\lambda_{IP}$  in practice? Besides, the best performance achieved by $L_{IP}$ is only marginally better to those without it.

3)	Some previous related works, such as InfoMin [Tian et al., 2020], are suggested to discussed and compared with more details. As far as I know, InfoMin analyzed influence of different view choices using mutual information theory. Therefore, it would be helpful to compare the similarities and differences between this work with it.

---

> ### Author Response · Authors · 2020-11-16
> **Response**
>
> [Remarks on controlled and uncontrolled experiments]
>
> We provide the experiments on CIFAR10 in Figure 3 (b), where the case when λ_IP=0 equals the standard uncontrolled experiment using the SimCLR [1] method. We use their released code [2] to replicate the result as well as adding the additional loss term L_IP. We also like to emphasize that MSCOCO is a large and diverse dataset with 330K images and around 5 captions per image. Then, we deploy the off-the-shelf SOTA baseline architectures (ResNet [3] and BERT [4]) on the MSCOCO dataset. The controlled experiments are performed for a better understanding of how different combinations of loss terms will affect the performance of the learned representations.
>
> [Remarks on λ_IP and L_IP]
>
> We thank the reviewer for raising the concern on incorporating the inverse predictive loss, and in what follows we provide an optimization perspective. We include the discussion in the revised manuscript and highlight it in red.
>
> One of the main goals of the experiment is to examine the setting when we learn the minimal and sufficient representations for self-supervision (see Definition 2 in the paper): minimize H(Z_X|S) under the constraint that I(Z_X; S) is maximized. However, this constrained optimization is not feasible when considering gradients methods in neural networks. Hence, we consider its practical Lagrangian Relaxation by a weighted combination between L_CL (or L_FP, representing I(Z_X; S)) and L_IP (representing H(Z_X|S)) with the λ_IP being the Lagrangian coefficient.
>
> The optimal λ_IP can be obtained by solving the Lagrangian dual, which depends on the parametrization of L_CL (or L_FP) and L_IP. Different parameterizations lead to different loss and gradient landscapes, and hence the optimal λ_IP differs across experiments. This conclusion is verified by the results presented in Figure 3(a), 3(b) and 4(b). Lastly, we point out that even not solving the Lagrangian dual, an empirical observation across experiments is that λ_IP which leads to the best performance is when the scale of L_IP is one-tenth to the scale of L_CL (or L_FP).
>
> [Discussion with InfoMin [5]]
>
> We have made the connection with our work with InfoMin [5] in the discussion of Theorem 1, and we are happy to provide a more detailed explanation of their similarities and differences. The following discussion will also be included in the revised manuscript and highlight in red.
>
> Both InfoMin [5] and our method suggest learning the representations that contain “not” too much information. In particular, InfoMin presents to augment the data (i.e., by constructing learnable data augmentations) such that the shared information between augmented variants is as minimal as possible, followed by the mutual information maximization between the learned features from the augmented variants. Our method instead considers standard augmentations (e.g., rotations and translations), followed by learning representations that contain only the shared information between the augmented variants of the data.
>
> [1] Chen et al., “A Simple Framework for Contrastive Learning of Visual Representations”, ICML 2020.
>
> [2] https://github.com/google-research/simclr
>
> [3] He et al., “Deep residual learning for image recognition”, CVPR 2016.
>
> [4] Devline et al., “Bert: Pre-training of deep bidirectional transformers for language understanding”, ACL 2019.
>
> [5] Tian et al., “What Makes for Good views for Contrastive Learning?”, NeurIPS 2020.

---

### Official Review · AnonReviewer3 · 2020-10-29
**Great theory but emprical evidence may need more clarification and improvement.**

**Rating:** 7
**Confidence:** 5

**Review:**

Summary:
Authors give an information-theoretical abstraction to represent various self-supervised learning methods and understand them better. Specifically, authors claim that self-supervised learning methods can extract task-relevant information and discard task-irrelevant information. Authors also do some controlled experiments and try to support their theoretical analysis.

Recommendation: I recommend to accept the paper (rating 7). I liked the abstraction proposed by authors and particularly liked the way authors set up the Definition 1 and analysis afterwards. Ratings can be improved further if authors can relate experimental setup more to the theory which I find slightly disconnected.

1) In assumption 1: I(X;T|S) <= \epslion. If task-relevant information lies mostly in the shared information between the input and the self-supervised signals then I(S;T|X) <= \epslion, should also be true, right?
2) Can authors give an intuitive proof sketch for Theorem 1?
3) In Proposition 1 and Theorem 3, it looks like d can be arbitrarily large and even scale with n. Is there any upper bound in d? Would this invalidate all claims?
4) In Figure 3 a and 3b, waht happens when you don’t use L_{CL} but use L_{FP} instead?
5) Figure 3 c is not clear. Why L_{CL} overfits and why does adding L_{FP} avoid that over-fitting? Why did you use \lambda_{FP} = 0.005?
6) From figure 3, It looks like L_{CL} is good enough if early stopping is used properly. How does one justify the use of FP and IP? One can also use some regularization along with CL (would the regularization act as “discarding  task-irrelevant  information”)?
7) How were hyper-parameters (learning rate, batch size, etc) chosen for all the experiments? Was proper hyper-parameter tuning done for all the methods? Can one achieve the same best results as in Figure 3a and 3b with proper tuning of method when \lambda_{IP} is zero?
8) It is not very convincing that controlled experiments in section 3 actually support Theorem 1 and 2. “Theorem 1 indicates that the self-supervised learned representations can extract almost as much task-relevant information as the supervised on” and “Theorem 2 indicates that a compression gap (i.e.,I(X;S|T)) exists when we discard the task-irrelevant information from the input”. Can authors clearly point out how conclusions from Figure 3 and Figure 4 are exactly supporting the respective statements of theorem 1 and 2 respectively?

---

> ### Author Response · Authors · 2020-11-16
> **Response**
>
> [I(X;T|S) and I(S;T|X)]
>
> We do not need the assumption of a small I(S;T|X). But if X and S are interchangeable, for example, X and S are different images with different augmentations on the same image, then we also assume a small I(S;T|X).
>
> [Proof sketch of Theorem 1]
>
> First, we note that the supervised learned representations are learning the representations that contain sufficient (and minimal) downstream information. Hence, I(Z_X^sup;T) = I(Z_X^sup_min;T) = I(X;T).
>
> Then, I(X;T) >= I(Z_X^ssl;T) by data processing inequality and I(Z_X^ssl;T) >= I(Z_X^ssl_min;T) by the fact that Z_X^ssl contains equal or more information than Z_X^ssl_min.
>
> Last, we discuss the inequality I(Z_X^ssl_min;T) >= I(X;T) - \epsilon_info.  Z_X^ssl_min is the representation that contains sufficient and minimal self-supervision (i.e., only the shared information between X and S). The downstream information that lies in X but not S is characterized as I(X;T|S), which is the difference between I(Z_X^ssl_min;T) and I(X;T).
> Specifically, I(Z_X^ssl_min;T) = I(X;T) - I(X;T|S). And by our assumption I(X;T|S) <= \epsilon_info, we conclude the proof.
>
>
> [d in Proposition 1 and Theorem 3]
>
> d can be seen as the parameters we used in our deep networks, which is fixed and does not scale with n.
>
> [Using L_FP instead of L_CL in Figure 3(a) and Figure 3(b)]
>
> The trend is similar that, with λ_IP being at a certain range, L_FP + λ_IP L_IP outperforms L_FP alone. Nonetheless, training L_FP requires much more training time than L_CL. In particular, in Figure 3 (c) we show that L_FP requires much more training epochs than L_CL to reach good performance. Hence, we choose to report the comparisons between L_CL + λ_IP L_IP and L_CL alone.
>
> [Overfitting between L_CL and L_FP, and λ_FP]
>
> We argue the reason that L_CL is more prone to overfitting than L_FP is that L_CL is easier to model than L_FP. On the one hand, L_CL aims to determine whether a pair of samples (Z_X, S) are sampled from the joint P_{Z_X, S} or the product of marginals P_{Z_X}P_S. The distinction between the joint and the product of marginals can be efficiently solved via recent tractable and scalable methods [1,2]. On the other hand, L_FP aims to reconstruct S from Z_X. Note that S stands for the self-supervised signal, which lies in a very high-dimensional space (e.g., S as an RGB image). Literature [3,4] has shown that reconstructing high-dimensional samples is challenging and requires a large number of training epochs to ensure a good reconstruction.
>
> The coefficient of λ_FP=0.005 is chosen to match the magnitude between L_CL and L_FP.
>
> [Usage of L_FP and L_IP other than L_CL]
>
> L_FP has benefits that 1) it does not contain a random sampling procedure from the product of marginals P_{Z_X}P_S, and 2) it does not require a large mini-batch size (e.g., n in equation 1). L_CL has benefits that 1) it does not need to reconstruct very high-dimensional samples, and 2) it takes smaller training epochs to reach good performance. Combining both of them empirically leads to good performance and is less prone to overfitting. Both L_FP and L_CL are aiming to extract task-relevant information, and we can decide to use which of them or both of them depends on the user setting.
>
> L_IP represents a realization of modeling H(Z_X|S), which aims to discard task-irrelevant information. As discussed in the related work section, another way to discard task-irrelevant information is minimizing I(Z_X ; X|S). Hence, realizations of modeling I(Z_X ; X|S) can also be regularizations along with L_CL and L_FP.
>
> [Hyper-parameters]
>
> The hyper-parameters of learning rates, batch size, and optimizer are fixed when comparing L_CL + λ_IP L_IP and L_CL alone. To be more specific, the only difference between them is  λ_IP. These hyperparameters are chosen to achieve the best performance on L_CL alone (λ_IP=0).
>
>
> [1] Poole et al., “On Variational Bounds of Mutual Information”, ICML 2019.
>
> [2] Tsai et al., “Neural Methods for Point-wise Dependency Estimation”, NeurIPS 2020.
>
> [3] Goodfellow et al., “Generative adversarial nets”, NeurIPS 2014.
>
> [4] Brock et al., “Large scale gan training for high fidelity natural image synthesis”, ICLR 2019.

---

> ### Author Response · Authors · 2020-11-16
> **Response (cont'd)**
>
> [Remarks on the experimental design]
>
> We thank the reviewer for suggesting a more clear statement on connecting theorems and experiments, which is definitely a plus to the paper. We include the following statement in the revised manuscript and highlight them in red.
>
> We present information inequalities in Theorems 1 and 2 regarding the amount of the task-relevant and the task-irrelevant information that will be extracted and discarded when learning self-supervised representations. Nonetheless, quantifying the information is notoriously hard and often leads to inaccurate quantifications in practice [5,6]. Not to mention the information we aim to quantify is the conditional information, which is believed to be even more challenging than quantifying the unconditional one [7]. To address this concern, we instead study the generalization error of the self-supervised learned representations, theoretically (Bayes error rate discussed in Section 2.4) and empirically (test performance discussed in Section 3).
>
> Another important aspect of the experimental design is examining equation (4), which can be viewed as a Lagrangian relaxation to learn representations that contain minimal and sufficient self-supervision (see Definition 2 in the paper): a weighted combination between I(Z_X; S) and -H(Z_X|S). In particular, the contrastive loss L_CL and the forward-predictive loss L_FP represent different realizations of modeling I(Z_X; S) and the inverse-predictive loss L_FP represents a realization of modeling -H(Z_X|S).
>
> As a summary, the first purpose of the experimental design is to empirically examine the generalization error of the self-supervised learned representations, as a surrogate inspection for Theorems 1 and 2. The second purpose is to study different realizations of learning representations that contain minimal and sufficient self-supervision (Definition 2 and equation (4)).
>
> [5] McAllester et al., “Formal Limitations on the Measurement of Mutual Information”, AISTATS 2020.
>
> [6] Song et al., “Understanding the Limitations of Variational Mutual Information Estimators”, ICLR 2020.
>
> [7] Poczos et al., “Nonparametric Estimation of Conditional Information and Divergences”, ICML 2012.

---

### Official Review · AnonReviewer2 · 2020-10-30
**This paper presented a novel theoretical framework to explore the self-supervised learning by information theory. The authors analyzed the self-supervised learning theoretically and proposed a composite objective including contrastive learning, forward predictive learning and inverse predictive learning. The paper is well written and easy to understand.**

**Rating:** 6
**Confidence:** 4

**Review:**

Strengths:
	Analyze the self-supervised learning methods from extracting task-relevant information and discarding task-irrelevant information from the input theoretically and clearly.
	Propose a new objective to discard task-irrelevant information explicitly. And it seems easy to combine with other SOTA self-supervised methods.
	Conduct experiments on visual setting and multi-modal setting.

Weakness:

	I like the idea of discarding the redundant task-irrelevant information to improve the self-supervised learning. However, the composite objective proposed in this paper seems just like a simple combination of three tasks, which is not strikingly novel.
	another concern of this paper is the lack of persuasive experiment results to prove the effectiveness of the proposed method. In fig.3, the improvements on two dataset are marginally, which can not convince me.
	The \lambda (λ_IP) in proposed objective function seems not robust to different datasets, which makes me doubt about the generalization of this method. I hope the authors provide more explanation about this.

---

> ### Author Response · Authors · 2020-11-16
> **Response**
>
> [Remarks on the composite objective and the role of λ_IP]
>
> We are happy to provide more discussions on the objective design and the role of λ_IP in the composite objective. In particular, we provide an optimization perspective to motivate the design and we will include the discussion in the revised manuscript and highlight it in red.
>
> The design of the composite objective is motivated by learning minimal and sufficient representations for self-supervision (see Definition 2 in the paper): minimize H(Z_X|S) under the constraint that I(Z_X; S) is maximized. However, this constrained optimization is not feasible when considering gradients methods in neural networks. Hence, we consider its practical Lagrangian Relaxation by a weighted combination between L_CL (or L_FP, representing I(Z_X; S)) and L_IP (representing H(Z_X|S)) with the λ_IP being the Lagrangian coefficient. In summary, the linear combination of three tasks is a practical relaxation of the original constrained optimization problem.
>
> Then, we discuss the choice of λ_IP. The optimal λ_IP can be obtained by solving the Lagrangian dual, which depends on the parametrization of L_CL (or L_FP) and L_IP. Different parameterizations lead to different loss and gradient landscapes, and hence the optimal λ_IP differs across experiments and datasets. This conclusion is verified by the results presented in Figure 3(a), 3(b) and 4(b). Lastly, we point out that even not solving the Lagrangian dual, an empirical observation across experiments is that λ_IP which leads to the best performance is when the scale of L_IP is one-tenth to the scale of L_CL (or L_FP).
>
> [Remarks on the experimental design]
>
> We thank the reviewer for suggesting a more clear statement on connecting theorems and experiments, which is definitely a plus to the paper. We include the following statement in the revised manuscript and highlight it in red.
>
> We present information inequalities in Theorems 1 and 2 regarding the amount of the task-relevant and the task-irrelevant information that will be extracted and discarded when learning self-supervised representations. Nonetheless, quantifying the information is notoriously hard and often leads to inaccurate quantifications in practice [1,2]. Not to mention the information we aim to quantify is the conditional information, which is believed to be even more challenging than quantifying the unconditional one [3]. To address this concern, we instead study the generalization error of the self-supervised learned representations, theoretically (Bayes error rate discussed in Section 2.4) and empirically (test performance discussed in Section 3).
>
> Another important aspect of the experimental design is examining equation (4), which can be viewed as a Lagrangian relaxation to learn representations that contain minimal and sufficient self-supervision (see Definition 2 in the paper): a weighted combination between I(Z_X; S) and -H(Z_X|S). In particular, the contrastive loss L_CL and the forward-predictive loss L_FP represent different realizations of modeling I(Z_X; S), and the inverse-predictive loss L_FP represents a realization of modeling -H(Z_X|S).
>
> As a summary, the first purpose of the experimental design is to empirically examine the generalization error of the self-supervised learned representations, as a surrogate inspection for Theorems 1 and 2. The second purpose is to study different realizations of learning representations that contain minimal and sufficient self-supervision (Definition 2 and equation (4)).
>
>
> [1] McAllester et al., “Formal Limitations on the Measurement of Mutual Information”, AISTATS 2020.
>
> [2] Song et al., “Understanding the Limitations of Variational Mutual Information Estimators”, ICLR 2020.
>
> [3] Poczos et al., “Nonparametric Estimation of Conditional Information and Divergences”, ICML 2012.

---

### Author Response · Authors · 2020-11-16
**General Response**

We thank all reviewers for the thoughtful feedback. We have addressed the concerns from the reviewers below and provided the suggested modifications in the revised manuscript and highlight them in red.

---

### Decision · Program_Chairs · 2021-01-07
**Final Decision**

**Decision:**

Accept (Poster)

**Comment:**

This paper received borderline reviews, but all lean toward acceptance.

The reviews highlighted strengths in the paper, citing that they liked the main idea and its mathematical treatment:
* R3: "I liked the abstraction proposed by authors and particularly liked the way authors set up the Definition 1 and analysis afterwards"
* R3 post-discussion: "I recommend accept because authors have a solid theory which would be useful for the self-supervised learning community."
* R4: "This work presents a very detailed theoretical analysis for self-supervised learning objectives. The idea of inverse predictive learning for filtering task irrelevant information is interesting."
* R2: "I like the idea of discarding the redundant task-irrelevant information to improve the self-supervised learning"

However, there was a consensus among reviewers that the experimental validation was weak, both in terms of not showing enough improvement on enough examples and in terms of studying the effect of certain hyperparameters:
* R2: "lack of persuasive experiment results to prove the effectiveness of the proposed method. In fig.3, the improvements on two dataset are marginally, which can not convince me. The \lambda (λ_IP) in proposed objective function seems not robust to different datasets, which makes me doubt about the generalization of this method."
* R3: "Ratings can be improved further if authors can relate experimental setup more to the theory which I find slightly disconnected"
* R3 post-discussion: "All reviewers have concerns about lack of solid experimental evidence [...]  I can not improve my score further because of weak experimental evidence."
* R1: "The experiments are conducted in a controlled way [...] Traditional uncontrolled experiments [...] are suggested."
* R4: "The variation in the performance shown in Figure 3 is very marginal. [...] Figure 5 a shows some results on Omniglot, but the improvement shown there is very marginal. [...]"
* R4: "weights required for inverse predictive learning in the loss formulation is not trivial. [...] Is there a simple way to determine this weights without exhaustive search on target dataset?"
* R4: "However, it is not clear from the experimental results if this is really effective."

The authors' revisions aim to improve the discussion of the $\lambda_\text{IP}$ parameter.

Given these experimental limitations, my recommendation is for acceptance but with a low confidence score.